# The unfolded protein response is required for dendrite morphogenesis

Xing Wei[1], Audrey S Howell[1], Xintong Dong[1], Caitlin A Taylor[1,2], Roshni C Cooper[1], Jianqi Zhang[3], Wei Zou[4], David R Sherwood[4], Kang Shen[1,2]*

[1]Department of Biology, Howard Hughes Medical Institute, Stanford University, Stanford, United States; [2]Neuroscience Program, Stanford University School of Medicine, Stanford, United States; [3]Division of Biostatistics, Department of Preventive Medicine, University of Southern California, Los Angeles, United States; [4]Department of Biology, Duke University, Durham, United States

**Abstract** Precise patterning of dendritic fields is essential for the formation and function of neuronal circuits. During development, dendrites acquire their morphology by exuberant branching. How neurons cope with the increased load of protein production required for this rapid growth is poorly understood. Here we show that the physiological unfolded protein response (UPR) is induced in the highly branched *Caenorhabditis elegans* sensory neuron PVD during dendrite morphogenesis. Perturbation of the IRE1 arm of the UPR pathway causes loss of dendritic branches, a phenotype that can be rescued by overexpression of the ER chaperone HSP-4 (a homolog of mammalian BiP/ grp78). Surprisingly, a single transmembrane leucine-rich repeat protein, DMA-1, plays a major role in the induction of the UPR and the dendritic phenotype in the UPR mutants. These findings reveal a significant role for the physiological UPR in the maintenance of ER homeostasis during morphogenesis of large dendritic arbors.

**\*For correspondence:** kangshen@stanford.edu

**Reviewing editor**: Graeme W Davis, University of California, San Francisco, United States

## Introduction

The organization of dendritic arbors is fundamental to the shape and connectivity of the nervous system (*Ramón y Cajal, 1911*; *Wassle et al., 1981*). Complex and type specific dendritic arbors are pivotal for many neurons to receive appropriate inputs from their receptive fields and to function properly in a neural circuit (*MacNeil and Masland, 1998*). During development, dendrites acquire their morphology by precisely regulated branch morphogenesis, which requires extracellular interactions and intracellular signaling pathways (*Jan and Jan, 2010*). For example, several diffusive or cell-surface molecules play instructive roles in guiding the growth and patterning of dendritic arbors. The diffusible chemoattractant Semaphorin 3A instructs the dendritic extension of cortical pyramidal neurons toward the pial surface (*Polleux et al., 2000*) while the graded expression of transmembrane Semaphorin 1A regulates the precise targeting of the dendrites of projection neurons in the *Drosophila* olfactory system (*Komiyama et al., 2007*). In the mammalian retina, a number of neuronal homotypic adhesion molecules, including Sdk1, Sdk2 and Cntn2, restrict dendritic arbors of amacrine cells and retinal ganglion cells in specific sublaminae in the inner plexiform layer (*Yamagata and Sanes, 2008*, *2012*; *Sanes and Zipursky, 2010*). Moreover, one common feature for dendrite development is that the sister branches from the same neuron avoid each other, while coexist with the branches of their neighboring neurons. This self-avoidance phenomenon has been elegantly elucidated by the function of two classes of highly diversified, contact-mediated repulsive molecules: Down syndrome cell adhesion molecules in *Drosophila* and protocadherins in vertebrates (*Schmucker et al., 2000*; *Wojtowicz et al., 2004*; *Matthews et al., 2007*; *Lefebvre et al., 2012*).

**eLife digest** The brain consists of billions of cells called neurons that can rapidly send and receive information. At one end of the neuron, branched structures called dendrites receive signals from other cells. The number of dendrites and the amount of branching vary in different types of neurons. These patterns are crucial for each neuron to receive the information it needs. Abnormalities in dendrites affect brain activity and are associated with several diseases in humans.

To make dendrites, the neuron needs to increase the amount of protein and other cell materials it produces. New proteins are made in a compartment called the endoplasmic reticulum and are folded into particular three-dimensional shapes with the help of chaperone proteins. These chaperones may be overwhelmed if protein production increases, leading to some proteins being folded incorrectly. This can activate a system called the unfolded protein response, which increases the number of chaperone proteins so that the proteins can be refolded correctly. However, it was not clear if neurons rely on the unfolded protein response, or another system, to cope with the increased levels of protein production needed to form complicated dendrite structures.

Wei et al. studied a type of neuron called PVD—which has an elaborate network of dendrites—in nematode worms. The experiments show that the unfolded protein response is activated in these neurons as the dendrites form. Mutant worms that were missing a protein called IRE1, which can activate the unfolded protein response, had dendrites with fewer branches than normal worms.

The experiments also show that a protein called DMA-1—which is required for dendrites to form—was not able to fold correctly in the mutant worms. As a result, this protein remained in the endoplasmic reticulum instead of moving to the surface of the cell where it is usually found. Wei et al.'s findings reveal that the unfolded protein response plays a major role in allowing cells to increase protein production as the dendrites form. The next challenge is to understand how neurons coordinate transcription and activation of the unfolded protein response.

These extrinsic cues must trigger intracellular signaling transduction that leads to cytoskeletal rearrangement as well as membrane biogenesis and trafficking (*Hanus and Ehlers, 2008*). For example, early endosome small G-protein RAB5 facilitates dendrite branching in *Drosophila* class IV da neurons (*Satoh et al., 2008*). Large cells with highly branched dendrites such as Purkinje cells accommodate the biosynthesis demand with a large soma containing extensive Golgi apparatus and abundant mitochondria (*Herndon, 1963*). Molecularly, the secretory pathway components including Sec23, Sar1, and Rab1 are particularly required for dendrite growth compared with axon development in the highly branched mammalian and *Drosophila* neurons (*Ye et al., 2007*). As part of the biosynthetic pathway, the production of membrane proteins requires protein folding in the endoplasmic reticulum (ER). It is currently unclear whether protein folding pathways play a role in the increased protein production required for dendrite development.

In the ER, a highly conserved protein quality control pathway, the unfolded protein response (UPR), maintains the ER homeostasis by adjusting the ER folding capacity upon detection of unfolded proteins (*Schroder and Kaufman, 2005*; *Ron and Walter, 2007*; *Walter and Ron, 2011*; *Worby and Dixon, 2014*). In higher eukaryotes, three proteins sense the ER stress and activate the UPR: the protein kinase (PKR)-like ER kinase (PERK), the activating transcription factor 6 (ATF6) and the inositol-requiring enzyme 1 (IRE1). Conserved in all eukaryotes, IRE1 contains an ER luminal domain, which is involved in the recognition of misfolded proteins in the ER, and cytoplasmic kinase and endoribonuclease domains, which can activate downstream pathways (*Credle et al., 2005*; *Gardner and Walter, 2011*) (*Figure 1—figure supplement 1A*). Activated IRE1 mediates the non-conventional splicing of an intron from the X box binding protein 1 (XBP1/HAC1) mRNA (*Cox and Walter, 1996*), and the IRE1-spliced XBP1 acts as a transcription factor to up-regulate the expression of ER chaperones such as BiP and other target genes to relieve the ER stress (*Travers et al., 2000*; *Lee et al., 2003*).

In the nematode *Caenorhabditis elegans*, the multidendritic polymodal nociceptive neuron PVD has an elaborate and organized dendritic arbor (*Figure 1A,B*). PVD's largely orthogonally arranged secondary (2°), tertiary (3°) and quaternary (4°) branches form repeated structural units resembling menorahs (*Smith et al., 2012*). During development, PVD grows its entire dendrite arbor that spans 800 μm along the body of the animal in just 24 hr (*Smith et al., 2010*), suggestive of a high level of

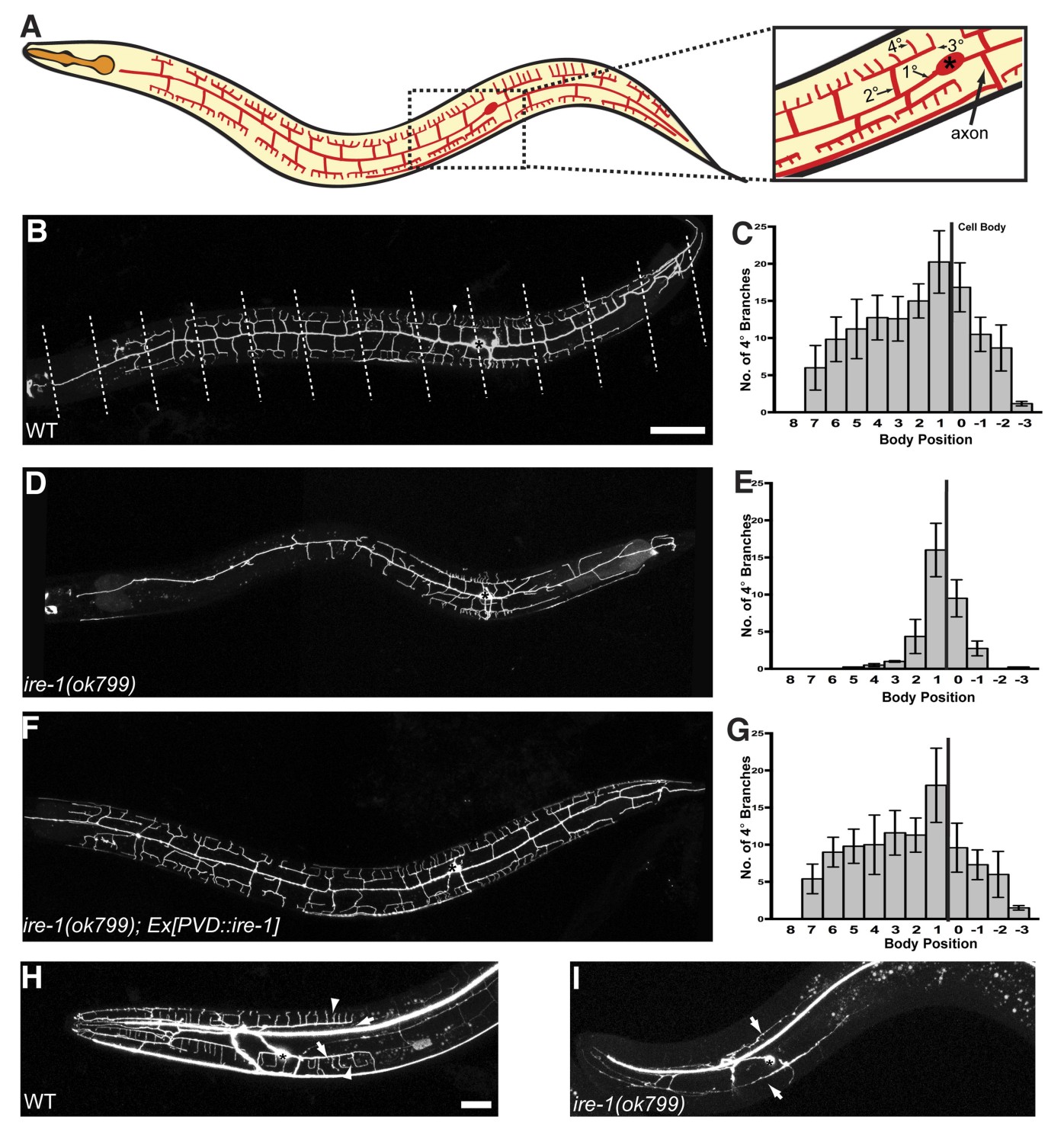

**Figure 1**. *ire-1* is required for PVD dendritic morphogenesis. (**A**) Cartoon showing the PVD dendritic arbor. The dash-boxed region is magnified to show the PVD soma (asterisk), axon, primary dendrite (1°), secondary dendrite (2°), tertiary dendrite (3°) and quaternary dendrite (4°). (**B**) Representative wild type (WT) dendritic morphology of PVD neuron expressing membrane associated mCherry (*wyIs581*). Starting from the cell body, the anterior and the posterior sections of the primary dendrite are divided into 8 and 4 equal segments, respectively, indicated by dashed lines. Anterior, left; dorsal, top. Asterisk, cell body; arrowhead, quaternary (4°) dendrite. Scale bars, 50 μm. (**C**) Quantification of the number of quaternary (4°) branches in each segment in WT. The position of cell body is indicated by the black line. Error bars show mean ± s.e.m., n = 10. (**D** to **G**) Defective PVD dendritic morphogenesis in *ire-1* (*ok799*) mutants (**D** and **E**) is rescued by expressing *ire-1* cell-autonomously (**F** and **G**). (**H** and **I**) Representative dendritic morphology of FLP neurons

*Figure 1. continued on next page*

*Figure 1. Continued*

labeled by cytoplasmic GFP in wild-type (**H**) and *ire-1* (*ok799*) mutants (**I**). Asterisks, cell bodies; arrows, secondary branches; arrowheads, tertiary branches. Scale bar, 20 μm.

The following figure supplements are available for figure 1:

**Figure supplement 1**. Schematic diagrams of IRE-1 dependent UPR pathway and of *C. elegans* IRE-1 protein showing three mutations.

**Figure supplement 2**. The *ire-1* mutants showing PVD dendritic morphogenesis defect.

**Figure supplement 3**. Compared with wild type animal (**A**), expressing *ire-1* cDNA (**B**) or spliced *xbp-1* cDNA (**C**) did not cause overbranching of PVD in wild type animals.

biosynthesis during the growth phase of this cell. The formation of PVD dendrites requires a single transmembrane leucine rich repeat (LRR) protein DMA-1, which acts cell autonomously in PVD to promote dendrite branching and stabilization (*Liu and Shen, 2012*). The elaborate dendritic branch pattern is instructed by hypodermal derived ligands SAX-7/L1CAM and MNR-1. Subcellularly localized stripes of SAX-7/L1CAM, together with MNR-1 form a tripartite receptor–ligand complex and guide the growth and branching of the PVD dendritic arbor (*Dong et al., 2013*; *Salzberg et al., 2013*).

Using two *ire-1* mutant alleles that we isolated from a dendrite morphology screen, we reveal that the physiological UPR is induced and required in the PVD neuron during dendrite morphogenesis. The IRE-1/XBP-1/BiP molecular cascade of the UPR pathway governs dendritic branching by regulating the folding and processing of DMA-1. Surprisingly, our evidence indicates that among many cell surface molecules required for dendrite formation, DMA-1 is largely responsible for the induction of the UPR.

## Results

### Loss of *ire-1* cause dendrite morphogenesis defects in highly branched neurons

We visualized the PVD neurons using a membrane associated mCherry or GFP marker expressed under the control of a cell-specific promoter (*ser2prom3::myr-mCherry* or *ser2prom3::myr-GFP*). From a forward genetic screen for mutations that alter the PVD dendritic morphology, we identified two loss-of-function mutations, *wy762* and *wy782*. Both alleles cause dramatic loss of dendritic branches, especially in the distal dendrites of PVD (*Figure 1—figure supplement 2*). Mapping and cloning of these two alleles showed that each allele contains a single point mutation in the *ire-1* (inositol-requiring 1 protein kinase) gene. In addition, a known null deletion allele of *ire-1*, *ok799* (*Henis-Korenblit et al., 2010*) showed indistinguishable phenotype in PVD compared with that of *wy762* and *wy782* (*Figure 1D*). The complexity of the menorahs nearest to the cell body appeared unaffected in these mutants (*Figure 1E*), as did the morphology of PVD axon (data not shown).

Interestingly, in the entire nervous system of *C. elegans*, the only other pair of highly branched neurons in the head region, FLP also showed severe dendritic arbor defects in *ire-1* mutants (*Figure 1H,I*). Other neurons with fewer dendritic or axonal branches such as IL2, VC and ADL did not show branching defects in *ire-1* mutants (data not shown). Together, these results suggest that *ire-1* is required for establishing highly branched dendrites.

To investigate where IRE-1 functions to regulate dendritic development, we generated transgenic mosaic animals. In the *ire-1* mutant background, expression of IRE-1 with a PVD-specific promoter (*ser2prom3*) fully restored the distal branch number and complexity of the whole dendritic arbor (*Figure 1F,G*) indicating that IRE-1 functions cell-autonomously in PVD to regulate dendrite morphogenesis. Expressing *ire-1* cDNA did not cause overbranching in wild-type animals (*Figure 1—figure supplement 3B*).

### Lack of folding capacity in the ER contributes to dendritic defect of PVD in *ire-1* mutants

IRE1 is conserved in all eukaryotes and contains an ER luminal domain for recognizing misfolded proteins in the ER, and a cytoplasmic kinase and an endoribonuclease domain, which lead to the

non-conventional cytoplasmic splicing of *xbp-1* (*Figure 1—figure supplement 1A*). One missense mutation (*wy782*) of *ire-1* is a substitution of a conserved residue in the kinase domain while another missense mutation (*wy762*) is a substitution of a conserved residue in the endoribonuclease domain (*Figure 1—figure supplement 1*), indicating both domains might be required for dendrite morphogenesis. Since these two domains are required for splicing of the *xbp-1* mRNA, we reasoned that the neurons should be able to bypass the requirement of IRE-1 if a spliced form *xbp-1* was provided in PVD. Consistent with this hypothesis, PVD-specific expression of spliced *xbp-1* cDNA in *ire-1* mutants rescued the loss of distal dendrite branches phenotype. In contrast, expression of unspliced *xbp-1* genomic DNA at the same concentration did not rescue branching defect (*Figure 2A–D*). Expressing spliced *xbp-1* cDNA did not cause overbranching in wild type animals (*Figure 1—figure supplement 3C*). These data offer compelling evidence that XBP-1 functions downstream of IRE-1 to establish complex dendritic arbor in PVD. Hence, the IRE-1 arm of the UPR pathway is likely involved in dendrite morphogenesis.

Because of the well-established role of the IRE-1/XBP-1 pathway in enhancing protein folding capacity in the ER, we hypothesized that IRE-1/XBP-1 upregulates specific ER chaperones to promote dendrite morphogenesis. We searched the PVD-specific gene profiling data (*Smith et al., 2010*) and found that two abundant ER chaperones of the Hsp70 family (homologous to mammalian grp78/BiP), HSP-3 and HSP-4, are enriched in PVD and therefore might be the targets of XBP-1 in PVD neurons (*Urano et al., 2002*). Consistent with this idea, overexpression of *hsp-4* in PVD restored normal dendritic branches in *ire-1* mutants (*Figure 2E,F*). However, overexpression of *hsp-3* or *daf-21* (a cytoplasmic chaperone of the Hsp90 family) did not rescue the phenotype (*Figure 2—figure supplement 1B,C*). Furthermore, *hsp-4* single mutant did not show the dendritic arbor defects (*Figure 2—figure supplement 1D*), indicating other ER chaperones or co-chaperones functioning in parallel with HSP-4. These results indicate that the dendritic defect in the *ire-1* mutants is likely due to lack of specific chaperones in the ER.

Importantly, *xbp-1* mutant (*Figure 2—figure supplement 2B*) did not show the dendritic arbor defects. This suggests that other pathways downstream of IRE-1 but independent of XBP-1 can play redundant roles in dendrite morphogenesis. During ER stress, Ire1 can promote the degradation of mRNAs encoding some ER proteins to maintain homeostasis through regulated Ire1-dependent decay (RIDD) (*Hollien and Weissman, 2006*; *Hollien et al., 2009*). The RIDD pathway has been shown to affect cell fate in various organisms, such as photoreceptor development in *Drosophila* (*Coelho et al., 2013*; *Maurel et al., 2014*). We next investigated whether the RIDD pathway functions in parallel with XBP-1 to regulate dendrite morphogenesis. mRNA degradation is initiated by internal cleavage mediated by RIDD, and the resulting RNA fragments would be subject to degradation by cytoplasmic 5′-3′ mRNA degradation machinery. However, all null mutants of the RIDD pathway components are lethal and difficult to examine dendrite phenotypes. Therefore, we used somatic clustered regularly interspaced short palindromic repeat (CRISPR) to create mosaic viable and conditional knock out of various genes (*Jinek et al., 2012*; *Cong et al., 2013*; *Shen et al., 2014*). Using this method, we found that in the *xbp-1* mutant background, somatic knockout of *xrn-1*, which encodes a 5′-3′ exoribonuclease and is a key component of the 5′-3′ mRNA degradation pathway (*Newbury and Woollard, 2004*), phenocopied the *ire-1* dendritic phenotype in PVD neurons in about 10% of the animals (*Figure 2—figure supplement 2C*). Somatic CRISPR is intrinsically mosaic and often generates low-penetrance phenotypes compared with viable null alleles. These results indicate that the RIDD pathway functions in parallel to the XBP-1 to regulate dendrite branching of PVD.

We also examined mutations in the other two arms of the UPR pathway, ATF-6 and PERK/PEK-1, and found that they did not show any dendrite morphogenesis phenotype in PVD (*Figure 2—figure supplement 3A,B*). However, *xbp-1 pek-1* double mutant showed a low-penetrance (about 25%) *ire-1*-like phenotype (*Figure 2—figure supplement 3C*). This suggests that the ER homeostasis mediated by other UPR pathways also contribute to dendrite morphogenesis.

## DMA-1 is a key target of the IRE-1 UPR pathway in PVD dendrite morphogenesis

We next asked which protein(s) are potential targets of the IRE-1 UPR pathway in PVD executing dendrite morphogenesis. The severe decrease of distal dendritic branches of PVD in *ire-1* mutants is reminiscent of *dma-1* mutants. DMA-1 is a single transmembrane leucine-rich repeat (LRR) protein

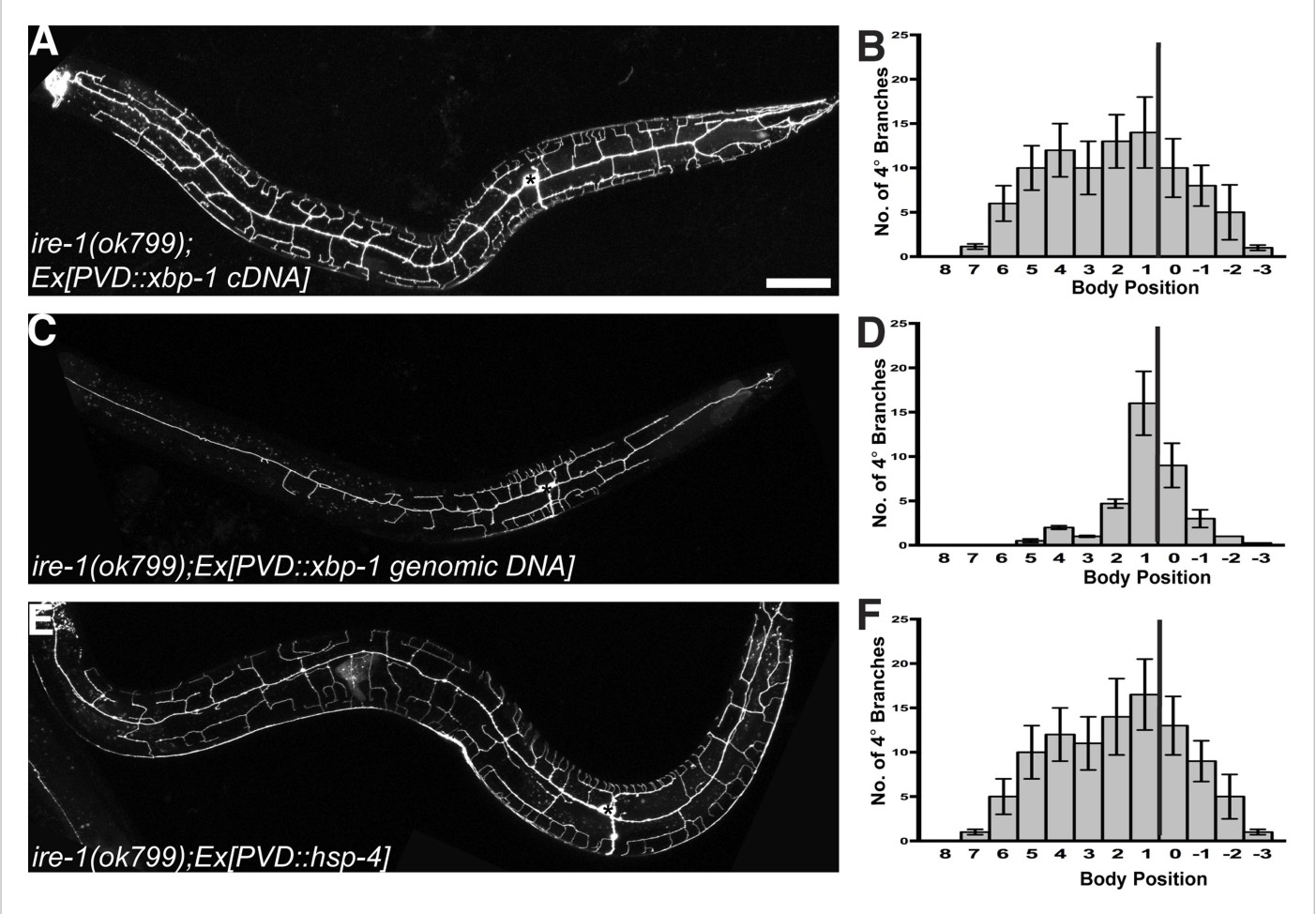

**Figure 2**. The UPR is required for PVD dendritic morphogenesis. Expressing *xbp-1* cDNA in PVD rescues the defective dendritic morphogenesis in *ire-1* mutants (**A** and **B**) while expressing *xbp-1* genomic DNA in PVD does not (**C** and **D**). (**E** and **F**) Expressing ER chaperone HSP-4 in PVD rescues the dendritic defect in *ire-1* mutants. Anterior, left; dorsal, top. Asterisks, cell bodies. Scale bar, 50 µm. Error bars show mean ± s.e.m., n = 10.

The following figure supplements are available for figure 2:

**Figure supplement 1**. Dendritic morphogenesis defects of PVD is likely due to lack of specific ER chaperones.

**Figure supplement 2**. The RIDD pathway in parallel with XBP-1 to regulate PVD dendritic arbor development.

**Figure supplement 3**. Other UPR arms also contribute to dendrite morphogenesis of PVD.

prominently expressed in PVD, and mutations in *dma-1* result in severely reduced dendritic branching and complexity (*Liu and Shen, 2012*) (*Figure 3—figure supplement 1A*). DMA-1 acts in PVD as a receptor to recognize the SAX-7/L1CAM and MNR-1 ligand complex in the surrounding skin cell to promote branching and precisely pattern the dendritic arbor (*Dong et al., 2013*; *Salzberg et al., 2013*). We reasoned that the folding of DMA-1 might require IRE-1. Consistent with this hypothesis, *ire-1 dma-1* double mutants showed a phenotype that is indistinguishable from the *dma-1* single mutant phenotype, suggesting that these two molecules function in the same genetic pathway in dendrite morphogenesis (*Figure 3—figure supplement 1B*). Furthermore, *hsp-4* overexpression in *ire-1 dma-1* double mutants was not able to rescue the dendritic arbor defect (*Figure 3—figure supplement 1C*), suggesting that DMA-1 might be a target of HSP-4.

As a single transmembrane protein, DMA-1 is synthesized in the ER and delivered to the plasma membrane through the secretory pathway. In wild type animals, GFP-tagged DMA-1 was detected on all the PVD dendritic processes and at the cortex of the cell body as diffusive fluorescence. In addition, numerous discrete intracellular puncta were found in the cell body and along the dendrites, which presumably represent the membrane trafficking organelles that carry DMA-1 (*Figure 3B*) (*Liu and Shen, 2012*). In *ire-1* mutants, the punctate DMA-1::GFP in the cell body was lost (*Figure 3E*). Instead, the somatic DMA-1::GFP in the *ire-1* mutants co-localized with an general ER marker, cytochrome b5 (cb5) (*Rolls et al., 2002*) (*Figure 3G*). Moreover, the diffuse DMA-1::GFP signal on the distal dendrites was dramatically reduced in the *ire-1* mutant while the signal on the proximal dendrites in *ire-1* mutants was the same as in wild type (*Figure 3H,J,K*). These observations suggest that DMA-1 is trapped in the ER and is not delivered to the distal dendrite plasma membrane, leading to the distal dendritic phenotype. Consistent with this hypothesis, overexpression of the ER chaperone HSP-4, restored the DMA-1::GFP subcellular localization in *ire-1* mutants to the normal distribution (*Figure 3J,K*). Taken together, these data suggest that ER chaperones such as HSP-4 help to fold DMA-1, which is required for the plasma membrane localization of DMA-1 and dendrite branching.

To further understand why the dendrite loss in the *ire-1* mutants was restricted to the distal dendrites, we investigated where the synthesis and folding of membrane proteins took place in PVD. This is an important question because the existence of local translation in dendrites might provide a source of DMA-1 production (*Holt and Schuman, 2013*; *Tom Dieck et al., 2014*). Since HSP-4 is capable of folding DMA-1, we first examined the subcellular localization of HSP-4 and found that HSP-4::GFP was exclusively localized in the PVD soma (*Figure 3—figure supplement 2A,D*), co-localizing with a rough ER marker TRAM (*Figure 3—figure supplement 2G–I*), HSP-4's ER localization pattern is consistent with the observation that its mammalian homolog BiP is localized in rough ER (*Bole et al., 1989*; *Lai et al., 2010*).These data suggests that the main protein synthesis and folding capacity for DMA-1 is likely in the cell body. In *ire-1* mutants, lack of the upregulation of *hsp-4* by spliced XBP-1 results in less DMA-1 in the secretory pathway and insufficient diffusion of DMA-1 to the distal region might be responsible for the specific loss of distal dendrites.

If the *ire-1* phenotype was the result of diminished DMA-1 levels in the distal dendrites, we reasoned two potential outcomes of DMA-1 overexpression in *ire-1* mutants. The increased expression of DMA-1 might reach the plasma membrane and rescue the *ire-1* phenotype. Alternatively, the DMA-1 overexpression might increase the protein-folding load and exacerbate the already strained protein folding machinery and lead to a more severe dendrite defect. Interestingly, we observed both effects: about 70% of animals showed efficient rescue of the dendritic arbor (*Figure 4B,D*), while about 25% of animals showed a more severe phenotype, with the loss of proximal branches in addition to the distal ones (*Figure 4C,D*). We hypothesized that in the absence of IRE-1, the remaining protein folding capacity is at a critical level where overexpression of DMA-1 can produce functional or misfolded proteins, possibly depending on the slightly variable levels of endogenous chaperones in individual animals (*Burga et al., 2011*). Consistent with the hypothesis, high level of chaperon HSP-4 expression together with DMA-1 decreased the percentage of *dma-1*-like phenotype, in a dose-dependent manner (*Figure 4D*). To further test this hypothesis, we separated the transgenic animals into the phenotypically rescued animals and the severely defective animals based on their dendrite morphology, we found that there was much less accumulation or aggregation of DMA-1::GFP in the PVD cell bodies with the rescued morphology compared to more severely defective animals (*Figure 4—figure supplement 1A–G*). Together, these rescuing results argue that the insufficient level of functional DMA-1 due to decreased protein folding capacity accounts for a large part of PVD dendritic defect in the *ire-1* mutants.

## The UPR activity in the PVD neuron is correlated with dendritic branching during development

We have shown that the UPR machinery is required for dendrite morphogenesis in PVD. However, it is not clear whether the dendritic branching activates the UPR in PVD during development. To answer this question, we designed a UPR activity reporter which contains the genomic fragment of *xbp-1* fused with a GFP in frame followed by an *SL2::mCherry* cassette (*Figure 5A*). Upon UPR activation, the intron in genomic *xbp-1* DNA will be spliced out by IRE-1, leading to the production of XBP-1::GFP (*Iwawaki et al., 2004*). The *SL2::mCherry* cassette permits the bicistronic expression of XBP-1::GFP

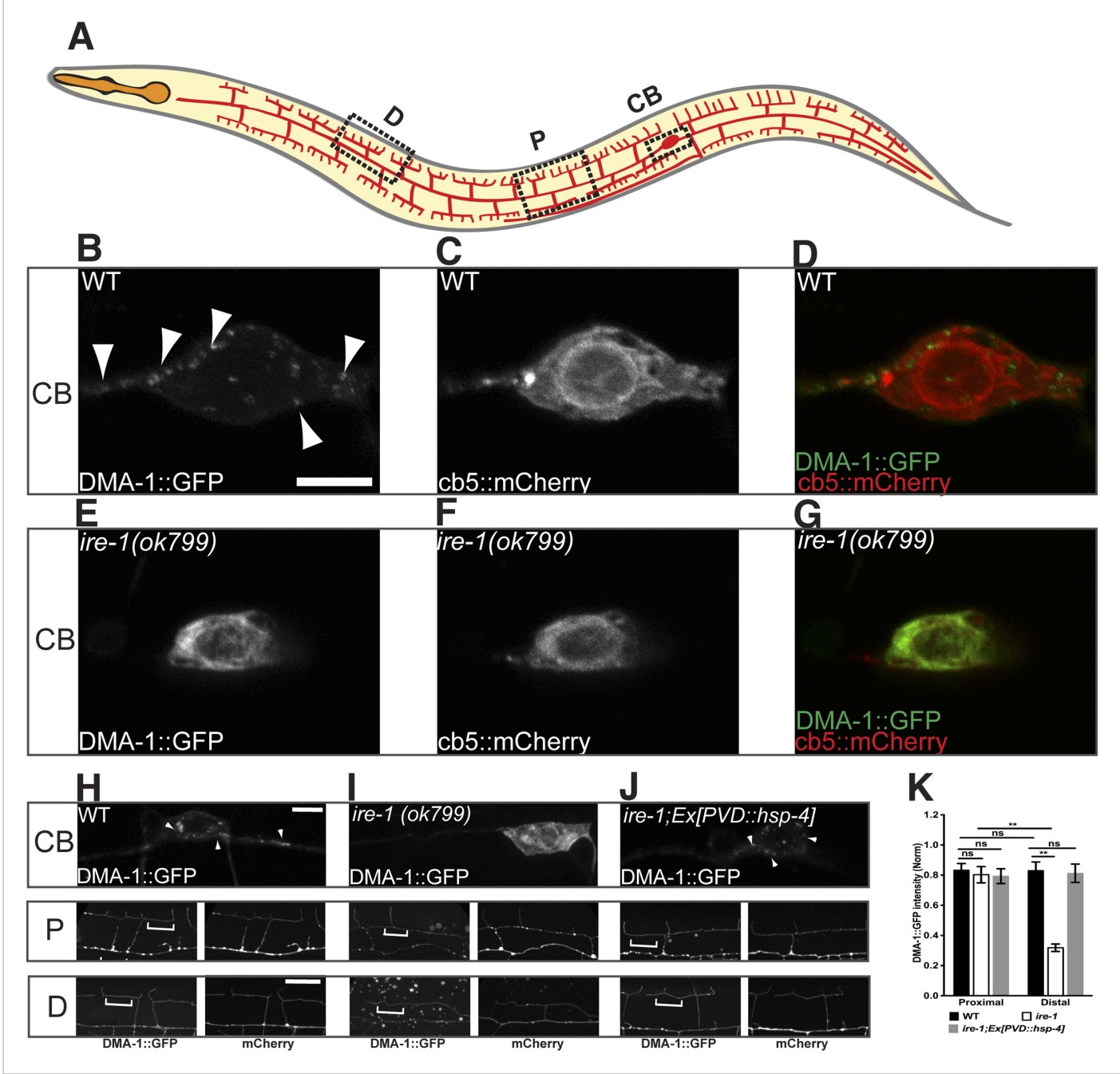

**Figure 3**. DMA-1 is stuck in the somatic ER in *ire-1* mutants. (**A**) Diagram of PVD in a young adult animal showing three representative subcellular regions: CB (cell body), P (proximal), and D (distal) dendrites. (**B** to **G**) Subcellular localization of DMA-1::GFP and general ER marker cb5::mCherry in PVD cell bodies in wild-type animals (**B** to **D**) and *ire-1* mutants (**E** to **G**). (**H** to **J**) DMA-1::GFP subcellular localization in WT (**H**), *ire-1* (**I**) and *ire-1* mutants expressing HSP-4 (**J**). Top panels: cell body; middle panels: proximal menorahs; bottom panels: distal menorahs. The morphology of the dendritic menorah is shown by cytoplasmic mCherry. Arrowheads, DMA-1::GFP puncta; brackets, tertiary branches (without puncta) used for measuring diffuse DMA-1:GFP. (**K**) Quantification of diffuse DMA-1:GFP (normalized to cytoplasmic mCherry) on tertiary branches in (**H** to **J**). Error bars show mean ± s.e.m., n = 50–60. ns, not significant; **p < 0.01 (two-way ANOVA and *post hoc* Sidak's multiple comparisons test). Scale bars, 5 μm.

The following figure supplements are available for figure 3:

**Figure supplement 1**. DMA-1 is required for PVD dendrite morphogenesis and may be the downstream of HSP-4.

**Figure supplement 2**. Subcellular localization of HSP-4::GFP, general ER marker cb5::mCherry and rough ER marker BFP::TRAM in PVD cell body (CB) region (**A** to **C** and **G** to **I**) and in distal (**D**) dendritic region (**D** to **F**) in wild type animals.

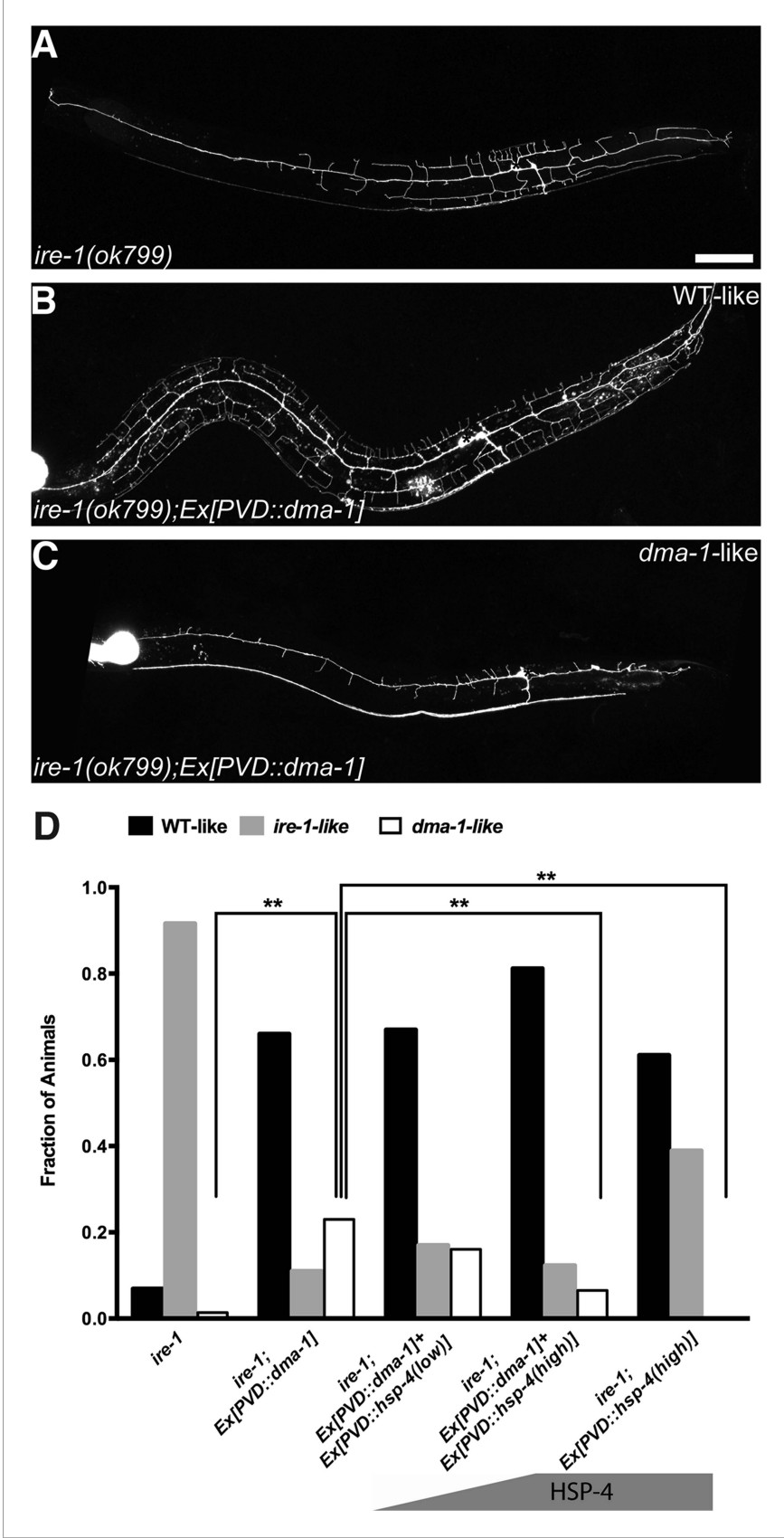

Figure 4. continued on next page

*Figure 4. Continued*

**Figure 4**. Overexpressing DMA-1 in *ire-1* mutants can either rescue dendritic defects or cause more severe dendrite branching defects. (**A**) Representative defective PVD dendritic arbor in *ire-1* mutants. (**B** and **C**) Overexpressing DMA-1 in PVD in *ire-1* mutants rescues the dendritic defect (WT-like) (**B**) or causes a more severe phenotype with fewer branches (*dma-1*-like) (**C**). Asterisks, cell bodies; Scale bar, 50 μm. (**D**) Proportions of different phenotypes in different transgenic rescue strains with overexpression of *dma-1* and/or supplementation of different doses of HSP-4 chaperon. n > 120. **p < 0.01, $\chi^2$ test with Sidak's multiple comparison correction.
The following figure supplement is available for figure 4:

**Figure supplement 1**. Accumulation or aggregation of DMA-1::GFP in the PVD cell bodies is tightly correlated PVD dendrite morphology.

and mCherry (*Spieth et al., 1993*), and its function is similar to the viral IRES sequence in mammalian system. The whole reporter is driven by a PVD specific promoter (*Pdes-2*). The XBP-1::GFP intensity indicates the endogenous UPR activity in PVD while the intensity of mCherry is used to normalize to transgene expression levels among different animals. Consistent with the requirement of IRE-1 to activate XBP-1, the XBP-1::GFP intensity in PVD neurons was diminished in *ire-1* mutants compared with wild-type animals (*Figure 6—figure supplement 1A,D*).

PVD neurons are derived postembryonically during the mid-L2 larval stage (*Sulston and Horvitz, 1977*), and starting from the late L2/early L3, 2° branches begin to form followed by extension of 3° branches in the L3 stage. Dendrite morphogenesis is completed in the early L4 stage after 4° branches have sprouted from the 3° branches to form a network of menorah-shaped processes (*Smith et al., 2010*). Using this PVD specific UPR activity reporter, we observed XBP-1::GFP in the nucleus of PVD starting at the L3 stage. The normalized XBP-1::GFP intensity increased between L3 and late L4 animals, coincidental with the stage of rapid dendrite branch addition. The XBP-1::GFP intensity subsequently decreased in mid-adult animals (*Figure 5B–K*). We verified this result by using another UPR activity reporter (*Phsp-4::HIS-24::GFP*). As an ER chaperone, HSP-4 is a transcriptional target of activated XBP-1 (*Calfon et al., 2002*; *Urano et al., 2002*), and its transcriptional level shows tight correlation with activation of the UPR with high sensitivity (*Iwawaki et al., 2004*). We used the *hsp-4* promoter to drive the expression of the *C. elegans* H1 histone, HIS-24 fused with GFP to detect the UPR activity in PVD neurons labeled with cytoplasmic mCherry. We found that the HIS-24::GFP signal became clearly detectable in L3 and further increased in L4 animals during which the menorahs form. The GFP fluorescence is dramatically downregulated in adult animals, (*Figure 5—figure supplemental 1*). Taken together, these results suggest that the UPR activity occurs most strongly during the time of PVD dendritic branching.

## DMA-1 is largely responsible for the activation of the UPR in PVD

The next question we wanted to address was how the UPR in PVD is induced during dendrite morphogenesis. The rapid dendritic growth of PVD requires high level of biosynthesis of plasma membrane proteins and efficient folding of them in the ER. PVDs are one of the only two pairs of highly branched neurons in *C. elegans*. Several transcription factors have been implicated in the cell fate determination of PVD. We considered two possibilities for the induction of the UPR activity in PVD neuron. In a 'top down' cell fate model, the enhanced UPR might be part of the cell fate decision controlled by transcription factors. Alternatively, the UPR might be induced because of the protein folding demand, in particular, maybe due to the translation of DMA-1, an essential membrane molecule for PVD dendrite branching (*Figure 6A*).

To distinguish these models, we first tested this PVD specific UPR activity reporter in *dma-1* knockout mutants and we found that the normalized XBP-1::GFP fluorescence level was significantly lower compared with wild type (*Figure 6B,E,K*), suggesting that a functional *dma-1* gene is required to turn on the UPR activity in PVD. Conversely, overexpression of *dma-1* cDNA in PVD leads to an increase of UPR activity (*Figure 6H,K*). Consistently, in *dma-1* mutants, another UPR activity reporter, *Phsp4::HIS-24::GFP* also showed dramatic decrease in PVD neurons (*Figure 6—figure supplement 2*).

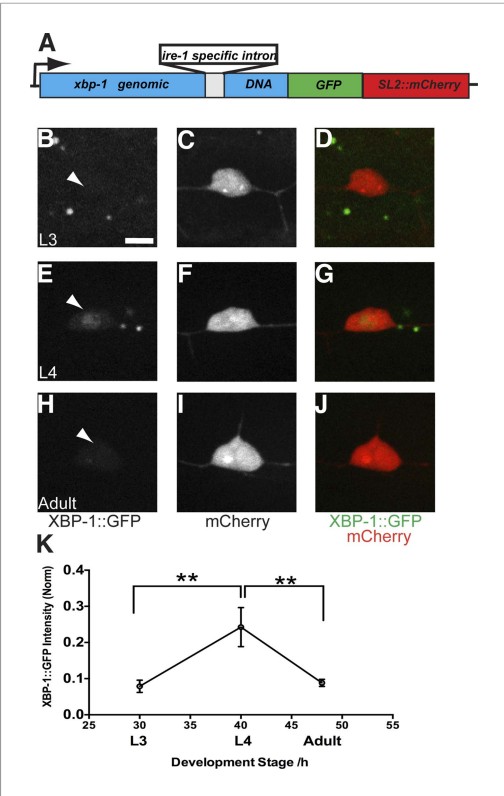

**Figure 5**. The UPR activity is correlated with dendritic branching during development in the PVD neuron. (**A**) Design of the PVD-specific UPR activity reporter. The *xbp-1* genomic DNA is fused with GFP followed by an *SL2::mCherry* cassette, which permits the bicistronic expression of XBP-1::GFP and mCherry. The reporter is driven by a PVD specific promoter. The XBP-1::GFP brightness indicates the UPR activity while the intensity of mCherry is used to normalize to transgene expression levels. (**B** to **J**) The PVD UPR activity in L3 stage (**B** to **D**), L4 stage (**E** to **G**) and adult stage (**H** to **J**). Arrowheads, nuclei of PVD. Scale bar, 5 μm. (**K**) Quantification of the normalized UPR activity in PVD in (**B** to **J**). Error bars show mean ± s.e.m. n = 30–50. **p < 0.01, Kruskal–Wallis one-way test and *post hoc* Dunn's test.

The following figure supplement is available for figure 5:

**Figure supplement 1**. The UPR activity is correlated with dendritic branching during development in the PVD neuron with another UPR reporter.

These surprising results argue that DMA-1 is necessary for UPR induction in PVD, despite the fact that there must be many membrane proteins necessary to build dendrites. To test whether DMA-1's role in UPR induction is specific, we asked if mutations in other membrane proteins required for PVD dendrite morphogenesis also result in a decrease in UPR activation. Deletion mutations in *kpc-1* (a Kex2/subtilisin-like propro-tein convertase and a Furin homolog) (*Schroeder et al., 2013*) and *hpo-30* (a claudin homolog) (*Smith et al., 2013*) cause severe reduction of PVD dendrites. While both of these gene products are processed in ER, neither mutation causes reduced UPR reporter activity in PVD (*Figure 6K* and *Figure 6—figure supplement 1G–L*). In addition, this reporter also showed activity in the unbranched neuron PVC, indicating that there might be UPR activity that is unrelated to branched dendrite morphogenesis. Neverthe-less, the *dma-1* mutation did not change the UPR activity in PVC (*Figure 6—figure supplement 3*). These results demonstrate that the activation of UPR in PVD specifically depends on DMA-1 production.

## Coexpression of HSP-4 and DMA-1 induces ectopic branches more efficiently

Surveying morphological phenotypes of other types of neurons, we found that the dendritic arbor defects in *ire-1* mutants were restricted to PVD and FLP the only two pairs of highly branched neurons in the *C. elegans* nervous system (*Figure 1I*). Coincidentally, only PVD and FLP showed sustained expression of DMA-1(*Liu and Shen, 2012*). These observations suggest that the establishment of a complex dendritic arbor not only requires instructive cell surface molecules but also physiological UPR to increase the protein folding capacity and maintain cellular homeosta-sis. Since PVD and FLP are also the largest neurons in worms with complicated dendritic arbor, we wondered if the UPR is particularly activated in these large cells. To directly test this idea, we examine the PVD morphology in *dpy-5 ire-1* double mutants. *dpy-5* mutants have reduced body length (about two third of that of wild type) due to bearing a deletion in the cuticle procollagen DPY-5 gene (*Thacker et al., 2006*) and correspondingly reduced PVD size (*Figure 7—figure supplement 1B*). Interestingly, the defective PVD phenotype of *ire*-1 was dramatically rescued with some animals showing wild type morphology (*Figure 7—figure supplement 1C–E*), indicating the UPR is particularly required for neurons with large and complicated dendritic arbors.

To further test this hypothesis, we determined the sufficiency of UPR activation to induce ectopic branches in neurons that normally do not branch extensively. The sensory neuron PDE shares the same lineage with PVD and does not express detectable levels of *dma-1*. The cell body of PDE is positioned

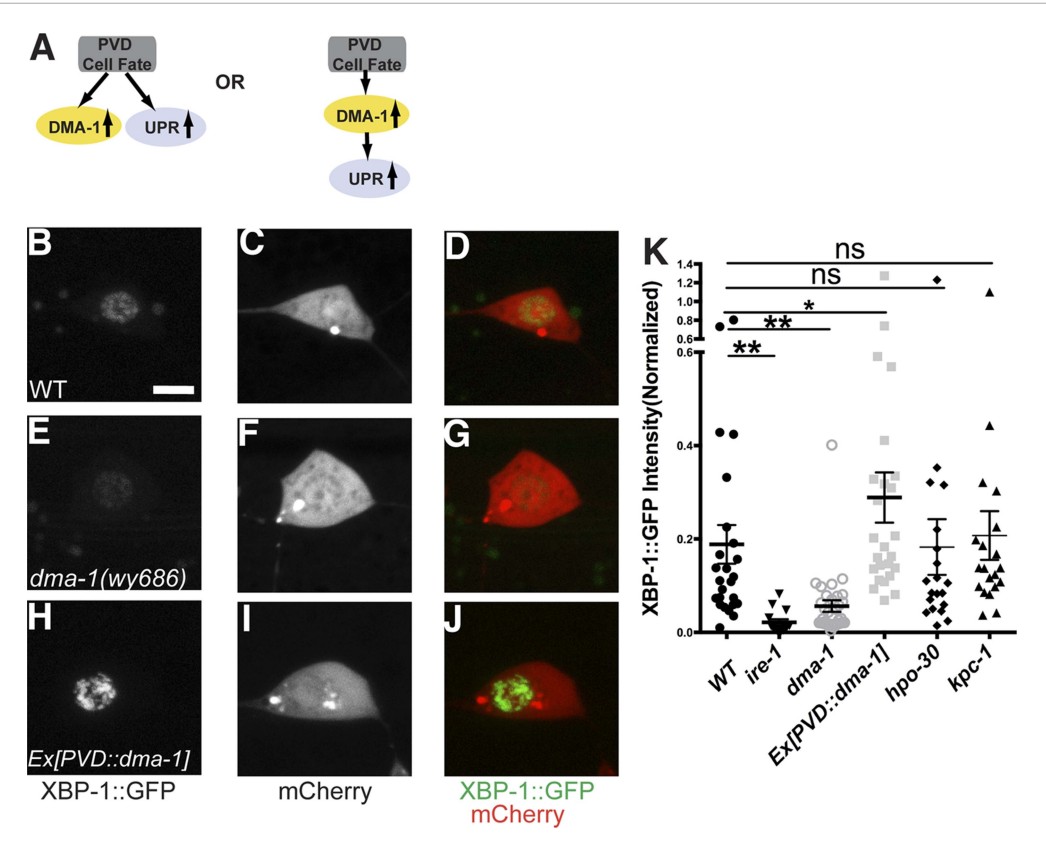

**Figure 6**. The induction of the UPR in PVD depends on the DMA-1. (**A**) Diagrams showing two possible models for the activation of the UPR in PVD. (**B** to **J**) The PVD UPR activity in WT (**B** to **D**), *dma-1* mutants (**E** to **G**) and WT with overexpession of *dma-1* (**H** to **J**). Scale bar, 5 μm. (**K**) Quantification of the normalized UPR activity in PVD in (**B** to **J**) and other mutants in (*Figure 6—figure supplement 1D–L*). Error bars show mean ± s.e.m. ns, not significant, **p < 0.01, *p < 0.05, Kruskal–Wallis one-way test and *post hoc* Dunn's test.

The following figure supplements are available for figure 6:

**Figure supplement 1**. The UPR activity in PVD does not depend on other known proteins that are processed in the ER and are required for PVD dendrite morphogenesis.

**Figure supplement 2**. Another UPR reporter in PVD also showed dramatic decrease in *dma-1* mutants.

**Figure supplement 3**. The UPR in unbranched PVC neurons does not depend on DMA-1.

---

close to the PVD's and has a single processes running adjacent to the PVD dendrites. Consequently, the extracellular environment for PDE including the molecular ligands for DMA-1 is similar to that of PVD (*Figure 7A,B*).

Overexpression of *dma-1* in PDE resulted in ectopic orthogonal branches that were similar to the PVD tertiary level branches (*Figure 7C*) (*Liu and Shen, 2012*). However, the low efficiency of ectopic branch induction (21% of animals bearing transgene) suggests there might be other intrinsic mechanisms that are necessary to establish exuberant branches. Notably, increasing the ER folding capacity by expressing HSP-4 together with DMA-1 in PDE induced ectopic branches more efficiently (47% of animals) while expressing other PVD-branching cell surface molecules such as HPO-30 with HSP-4 did not induce any ectopic branches (*Figure 7D–G*). Further, these ectopic branches were more branched and significantly longer than overexpressing DMA-1 alone (*Figure 7H*), and the *Phsp-4::HIS-24::gfp* reporter also showed increased UPR activity in PVD (*Figure 7—figure supplement 2*). These data support our model that the UPR is required for highly branched dendrites.

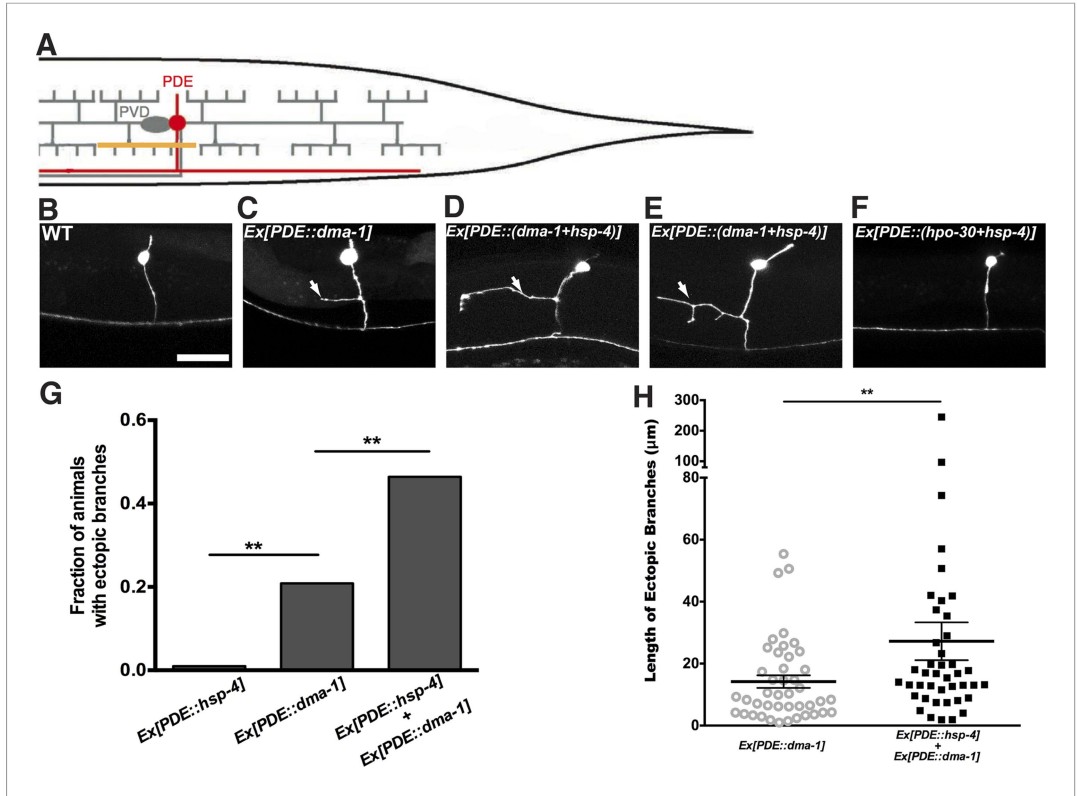

**Figure 7**. Expression of *dma-1* and *hsp-4* together in the morphologically simple PDE neurons can induce ectopic branching more dramatically. (**A**) Diagram showing the PDE neuron (in red) which is located close to the PVD cell body and has a simple processes running adjacent to the PVD dendrites. Orthogonal ectopic branching (in orange) at a stereotyped position in the PDE commissure, reminiscent in location and direction to PVD tertiary branching. (**B** to **D**) Representative dendritic morphology of PDE neurons expressing cytoplasmic GFP in wild-type (**B**), a strain with expression of *dma-1* (**C**) and a strain with co-expression of *dma-1* and *hsp-4* in PDE (**D**). Scale bar, 20 μm. Arrows, orthogonal ectopic branches. (**E**) More dramatic ectopic branching in the same strain co-expressing *dma-1* and *hsp-4* in PDE. (**F**) No ectopic branching in the same strain co-expressing *hpo-30* and *hsp-4* in PDE. (**G**) Percentages of PDE ectopic branching in strains expressing *hsp-4* only, *dma-1* only, and *dma-1* with *hsp-4*. n > 100. **p < 0.01, $\chi^2$ test with Sidak correction. (**H**) Length of PDE ectopic branches in the strains expressing *dma-1* only, and *dma-1* with *hsp-4*. **p < 0.01, Mann Whitney U-test.

The following figure supplements are available for figure 7:

**Figure supplement 1**. The defective PVD phenotype of *ire*-1 is suppressed by reduced size of PVD dendritic arbor.

**Figure supplement 2**. Overexpression of *dma-1* in PDE induces ectopic branches and increases UPR activity.

## Discussion

### The unfolded protein response is an intrinsic requirement for highly branched neurons

Conserved in all eukaryotes, the UPR pathway plays significant roles in dealing with cellular stress and balancing homeostasis and apoptosis (*Walter and Ron, 2011*). Failure to mitigate the ER stress and reestablish homeostasis correlates with cell death, playing a central role in numerous human diseases such as pancreatic β-cell loss in diabetes (*Fonseca et al., 2011*), retinal degeneration triggered by misfolded proteins in retinal dystrophies (*Lin and Lavail, 2010*) and dopaminergic neuron degeneration in Parkinson disease models (*Valdes et al., 2014*). In addition, under other biological conditions involving intense ER functions such as viral infection (*Dimcheff et al., 2003*) or pathogen

defense (*Richardson et al., 2010*), the UPR is activated to relieve the ER stress. In the nervous system, several reports indicate normal regulatory roles of the IRE-1. For example, IRE-1 was shown to be involved in trafficking cell surface molecules such as AMPA receptor in cultured cells (*Vandenberghe et al., 2005*) or glutamate receptor GLR-1 in *C. elegans* interneurons (*Shim et al., 2004*), and rhodopsins in *Drosophila* photoreceptors (*Coelho et al., 2013*).

However, to the best of our knowledge, the role of the UPR pathway in the development of the nervous system is poorly understood except in a few isolated cases. For example, a different arm of the UPR, PERK1, was recently shown to be required for the olfactory receptor choice in mammalian olfactory sensory neurons through a feedback loop (*Dalton et al., 2013*). Our results on the IRE-1/XBP-1/BiP/DMA-1 molecular cascade regulating dendrite morphogenesis in PVD neurons provide another clear example that UPR is directly involved in the development of neuronal cell morphology. More importantly, these results provide a link between dendrite morphogenesis and cellular homeostasis. First, only highly branched neurons such as PVD and FLP require the UPR to establish their dendritic arbors; second, the UPR activity in PVD is correlated with the development of dendrites; finally the induction of UPR is dependent on the expression of a pivotal molecule for dendritic branching DMA-1 in a homeostatic manner. Thus, in addition to the instructive extracellular cues required for complex branch formation and guidance, intrinsic mechanisms are also required.

## The UPR and secretory pathways are critical for dendrite morphogenesis

The secretory and endocytic pathways constitute the main membrane trafficking pathway in cell (*Sallese et al., 2006*; *Brandizzi and Barlowe, 2013*). In order to form dendrites, membrane and transmembrane proteins need to be synthesized and delivered to the growing dendritic arbor. It is therefore not surprising that molecular components, in these pathways such as the early endosome protein RAB-5 are involved in dendrite morphogenesis (*Satoh et al., 2008*). Interestingly, one previous study showed that secretory pathway mutants preferentially alter dendrite morphology and not axon extension (*Ye et al., 2007*). Similarly, we found that the dendrite but not axon morphogenesis is specifically compromised in the *ire-1* mutants, suggesting that specific molecular program and membrane trafficking pathways are required for dendrite development. Thus, our studies add weight to the idea that different molecular and trafficking pathways are utilized during dendrite morphogenesis and axon extension.

## The UPR for dendrite morphogenesis is largely triggered by A single transmemebrane protein

The physiological functions of the UPR have been best demonstrated in highly differentiated cells that produce specific types of proteins in large amounts. One best-characterized example is the requirement of IRE1 and XBP1 for differentiation of B cells into plasma cells, where the UPR is activated to accommodate the secretion of large amounts of immunoglobulins (*Reimold et al., 2001*). In these cells, the ER and secretory system are highly specialized for antibody biosynthesis, which accounts for half of the total protein production in these cells (*Askonas, 1975*). In Ig heavy chain knockout mice, the UPR activity was diminished in B cells, indicating that the production of immunoglobulins in B cells is required for induction of the UPR (*Iwakoshi et al., 2003*). Another example is in the mammalian olfactory sensory neurons. Olfactory receptor (OR) genes are among the most highly transcribed G-protein-coupled receptors (GPCRs) in these neurons and trigger the UPR feedback loop required for specific OR choice (*Dalton et al., 2013*).

In contrast to these specialized cell types, a large number of diverse proteins and lipids are required in a developing neuron to establish its dendritic arbor. Many of these proteins are folded and processed in the ER. Surprisingly, our results suggest that among these proteins, a single transmembrane protein, DMA-1, appears to be largely responsible for the activation of the UPR pathway in PVD during dendritic development. The characteristics of the DMA-1-like LRR proteins including their non-globular flexible solenoid like structure, repetitive amino acid sequences and high content of hydrophobic leucines, might make them particularly challenging to fold and assemble properly (*Freiberg et al., 2004*). These characteristics may trigger the UPR in these cells and thus make this system required for dendrite morphogenesis. Other evidence also supports the notion that specific proteins have higher folding demands. For example, although IRE-1 has been shown to function in the normal secretory pathway (*Safra et al., 2013*), the protein trafficking defect in *ire-1*

mutants is not general. It has been shown that for several different membrane-spanning proteins including Golgi-resident mannosidase, a TWK-type potassium channel, a single transmembrane synaptic vesicle protein synaptobrevin and a vesicular monoamine transporter CAT-1, their subcellular localizations in *ire-1* mutants were unaffected in interneurons (*Shim et al., 2004*). These results suggest that DMA-1 is specifically regulated by the UPR pathway in the PVD neuron.

Together, these findings indicate that certain proteins are intrinsically more challenging for folding and specific cell types have to employ the UPR pathway to accommodate the influx of these proteins and maintain homeostasis in the ER.

## Materials and methods

### Strains and genetics

Strains were grown at 20°C on NGM agar plates seeded with *Escherichia coli* OP50 except the UPR mutants and UPR reporter strains growing at 16°C. The wild-type strain was *C. elegans* N2 Bristol. The following mutant alleles and transgenes were used in this study:

LGI: *dma-1(wy686), kpc-1(gk8) dpy-5 (e907)*; LGII: *ire-1(ok799), hsp-4(gk514)*; LGIII: *xbp-1(tm2482) wyIs592 [ser2prom3::myrGFP, odr-1::dsRed]*; LGIV: *wyIs581[ser2prom3::myr-mCherry, odr-1::dsRed]*; LGV: *hpo-30(ok2047)*; LGX: *atf-6(ok551), pek-1(ok275), qyIs369[ser2prom3::dma-1::GFP,unc-119+])*, *wyIs378[ser2prom3::myrGFP, Prab3::mCherry, odr-1::dsRed]*.

### Isolation and mapping of mutants

The *wy762* and *wy782* alleles were isolated from an F2 semiclonal screen of 3000 haploid genomes in the strain containing *wyIs378* (*Dong et al., 2013*). Based on SNIP-SNP mapping and whole genome sequencing (*Sarin et al., 2008*), we got the missense mutation information on *ire-1* locus and verified by Sanger sequencing and complementation test with the null allele.

### Molecular cloning

Expression clones were made in the pSM vector, a derivative of pPD49.26 (A Fire) with extra cloning sites (a kind gift from S McCarroll and CI Bargmann). The *ser2prom3* (PVD) and *Pdat-1* (PDE) promoters were used for cell-specific expression. cDNAs of *ire-1*, *xbp-1*(long isoform), *hsp-3* and *his-24* were amplified from cDNA library while genomic DNAs of *xbp-1*, *hsp-4*, *cb5* (C31E10.7) and *tram-1* were amplified from genomic templates. The XBP-1 UPR reporter construct was driven by *Pdes-2* (PVD), containing *xbp-1* genomic DNA fused with GFPnovo2 (*Arakawa et al., 2008*) followed by *gpd-2 SL2::mCherry* (from pBALU12) (*Tursun et al., 2009*). For *hsp-4* transcriptional activity reporter, the 1.1 kb 5′ upstream of *hsp-4* ATG was cloned, driving HIS-24 fused with GFPnovo2. For HSP-4:: GFP, GFPnovo2 was inserted right before the C-terminus HDEL sequence of genomic HSP-4. For somatic CRISPR, two DNA templates of *xrn-1* sgRNA were 5′- GATATCGCTCCGATGTCCAT-3′ and 5′- AACGTGACGTCATCGTCATT-3′, under the control of *U6* promoter as in (*Chen et al., 2013*).

### Germline transformation

The transgenic extrachromosomal arrays were generated via injection using standard microinjection techniques (*Mello and Fire, 1995*).

For rescue experiments, *wyEx7329[ser2prom3::ire-1 (40 ng/μl), pBluescript (60 ng/μl), odr-1::dsRed (90 ng/μl)]*; *wyEx7332[ser2prom3::xbp-1(cDNA) (20 ng/μl), pBluescript (60 ng/μl), odr-1::dsRed(90 ng/μl)]*; *wyEx6502[ser2prom3::xbp-1(genomic DNA) (40 ng/μl), pBluescript (60 ng/μl), odr-1::dsRed(90 ng/μl)]*; *wyEx6816[ser2prom3::hsp-4 (50 ng/μl), pBluescript (60 ng/μl), odr-1::dsRed(90 ng/μl)]*; *wyEx7333 [ser2prom3::hsp-3 (50 ng/μl), pBluescript (60 ng/μl), odr-1::dsRed(90 ng/μl)]*; *wyEx7335[ser2prom3::daf-21 (50 ng/μl), pBluescript (60 ng/μl), odr-1::dsRed(90 ng/μl)]*.

For ER markers and chaperone co-labeling, *wyEx8074[ser2prom3::cb5::mCherry PCR fusion product (20 ng/μl), ser2prom3::hsp-4::GFPnovo2::HDEL (10 ng/μl), pBluescript (30 ng/μl), odr-1:: dsRed(90 ng/μl)]*; *wyEx8075[Pdes-2::tagBFP::TRAM (15 ng/μl), ser2prom3::hsp-4::GFPnovo2::HDEL (10 ng/μl), pBluescript (30 ng/μl), odr-1::dsRed(90 ng/μl)]*.

For DMA-1 overexpression experiment, in *wyIs581* background, *wyEx7338[ser2prom3::dma-1:: GFP (50 ng/μl), pBluescript (60 ng/μl), Pmyo-2::mCherry(1.5 ng/μl)]*;

For HSP-4 dose-dependent rescue experiments, with wyEx7338 and wyIs581, wyEx7859 [ser2prom3::hsp-4 (30 ng/µl), pBluescript (60 ng/µl), odr-1::dsRed(60 ng/µl), pBluescript (60 ng/µl)]; wyEx7770[ser2prom3::hsp-4 (60 ng/µl), pBluescript (30 ng/µl), odr-1::dsRed(60 ng/µl)].

For the UPR activity reporter, wyEx6766[Pdes-2::xbp-1(genomic)::GFPnovo2::SL2-mCherry (80 ng), Punc-122::dsRed(30 ng/µl), pBluescript (30 ng/µl)]; wyEx6812[ser2prom3::dma-1 (50 ng/µl), odr-1:: dsRed(60 ng/µl), pBluescript (60 ng/µl)] For UPR activation experiment with hsp-4 transcriptional reporter, with wyIs581, wyEx7820[Phsp-4::HIS-24::GFPnovo2 (20 ng/µl), pBluescript (60 ng/µl), odr-1:: dsRed(70 ng/µl)].

For somatic CRISPR, in xbp-1 (tm2482) background, wyEx7862[Phsp-16.2::Cas9 (50 ng/µl), PU6:: xrn-1-sgRNA1 temp (30 ng/µl), PU6::xrn-1-sgRNA2 temp (30 ng/µl),odr-1::GFP(40 ng/µl)].

For PDE ectopic branching experiments, wyEx7035 [Pdat-1::hsp-4 (40 ng/µl), Pdat-1::GFP (20 ng/µl), odr-1::dsRed(60 ng/µl), pBluescript (30 ng/µl)] injected into wyEx4287 strain with overexpression of dma-1 in PDE (Liu and Shen, 2012); wyEx8063[Pdat-1::GFP (20 ng/µl), Pdat-1::hsp-4 (40 ng/µl), Pdat-1::hpo-30 (30 ng/µl), odr-1::dsRed (90 ng/µl)];

For PDE UPR reporter experiments, wyEx8049[Phsp-4::HIS-24::GFPnovo2 (20 ng/µl), Pdat-1:: mCherry (2 ng/µl), pBluescript (60 ng/µl), odr-1::dsRed(70 ng/µl)]; then use this line to ectopic express dma-1 in PDE, wyEx8065 [Pdat-1::dma-1::BFP (50 ng/µl), Pdat-1::mCherry (20 ng/µl), pBluescript (30 ng/µl), odr-1::GFP(20 ng/µl)].

wyEx4280 was used for FLP labeling (Liu and Shen, 2012).

## Somatic xrn-1 CRISPR

Following the protocol in (Shen et al., 2014) with some modifications, we first synchronized the culture by allowing 100–150 adult worms containing transgenic arrays (raised at 20°C) to lay eggs for 3 hr on seeded NGM plates. The eggs were heat-shocked at 33°C for 2 hr and then shifted to 20°C. After 60 hr, the PVD morphology was checked at the young adult stage.

## Microscopy

Images of fluorescently tagged fusion proteins were captured in live C. elegans using Plan-Apochromat 40×/1.3NA objective for whole PVD morphology and 63×/1.4NA for subcellular localization of fluorescent proteins on a Zeiss LSM710 confocal microscope (Carl Zeiss, Germany). Animals were immobilized on 2% agarose pad using 10 mM levamisole (Sigma–Aldrich, St. Louis, MO) and oriented anterior to the left and dorsal up. Z-stacks were collected and the maximum intensity projection was used for additional analysis. For analyzing DMA-1::GFP intensity on tertiary dendrites (middle and bottom panels in Figure 3H–J), XBP-1::GFP intensity during development (Figure 5B–J) and HIS-24:: GFP UPR activity reporter (Figure 5—figure supplement 1, Figure 6—figure supplement 2 and Figure 7—figure supplement 2), images were acquired using a Zeiss Axio Observer Z1 microscope equipped with a Plan-Apochromat 63×/1.4NA objective, Yokogawa spinning disk head (Japan), 488 nm and 561 nm diode lasers (Coherent, Santa Clara, CA), and a Hamamatsu ImagEm EMCCD camera (Japan) driven by MetaMorph (Molecular Devices, Sunnyvale, CA).

## Images quantification

For 4° dendrite number counting, two PVD images (labeled by wyIs581) from late L4 or young adults were stitched together in Adobe Photoshop (San Jose, CA). The general shape and location of the primary dendrite (the 'backbone') was recognized by a model-based neurite fiber tracing method (Peng et al., 2008). Then the length of primary dendrite was determined by tracing the backbone and calculating the distance between adjacent identified pixels. Finally, the anterior part from cell body was divided into 8 equal segments while the posterior part was divided into 4 equal segments (written in custom Matlab scripts (Mathworks, Natick, MA)). It should be noted that the length of each anterior segment is not equal to each posterior segment. The numbers of 4° dendrites whose secondary dendrites grew in each segment were counted manually.

For 2° dendrite number counting, two PVD images (labeled by wyIs581 or wyIs592) from late L4 or young adults were stitched together in Photoshop. Then the length of primary dendrite was determined manually by tracing the backbone and calculating the distance between adjacent identified pixels. The 2° dendrite number in each animal was counted manually, and this number was divided by the length of PVD primary dendrite (per 100 µm).

For measuring DMA-1::GFP intensity on 3° dendrites, we chose menorahs around the vulva region as 'Proximal' to avoid numerous puncta in dendrites close to cell body and chose menorahs around the middle point of anterior primary dendrite as 'Distal' to make sure we could get T-like branches in this region in *ire-1* mutants. Two channel images were combined together by ImageJ (Wayne Rasband), and a 2-pixel wide line was drawn along the tertiary branches (avoiding obvious puncta) and then the mean intensity values of two separated channels along this line were measured. After subtracting the background signal, the DMA-1::GFP signal was normalized to cytoplasmic mCherry. 3–5 tertiary branches were chosen for each spinning-disk image.

To quantify the UPR activity, for different genotypes, the XBP-1::GFP or HIS-24:GFP mean intensity in the nucleus (after background subtraction) measured with ImageJ was normalized to the mean intensity of cytoplasmic mCherry in the same region using custom written Python (Python Software Foundation, Beaverton, OR) scripts. HIS-24::GFP intensity (*Figure 5—figure supplement 1* and *Figure 7—figure supplement 2*) was measured and quantified without normalization to cytoplasmic mCherry.

All custom matlab and Python codes are provided in the *Source code 1*.

## Statistical analysis

In comparisons of measurements such as fluorescence intensity or length of ectopic branches, we first tested for normality using a D'Agostino-Pearson test (alpha = 0.05). For data sets with normal distribution, we applied a two-tailed Student's t test for comparisons of two groups (*Figure 6—figure supplement 3G* and *Figure 4—figure supplement 1G*). Comparisons involving multiple groups with multiple factors used two-way ANOVA and *post hoc* Sidak's multiple comparisons test (*Figure 3K*). For data sets without normal distribution, we applied a two-tailed Mann–Whitney *U-test* for comparisons of two groups (*Figure 7H* and *Figure 7—figure supplement 2G*). Comparisons involving multiple groups used Kruskal–Wallis one-way test and *post hoc* Dunn's test (*Figures 5K, 6K*, *Figure 5—figure supplement 1J* and *Figure 6—figure supplement 2J*). To compare variables such as proportions we used $\chi^2$ test with Sidak correction for multiple comparisons (*Figure 4D* and *Figure 7F*). All statistical tests were performed in Graphpad Prism (San Diego, CA) or in R (R Development Core Team).

## Acknowledgements

This work was supported by the Howard Hughes Medical Institute and by the NIH (1R01NS082208). We thank Shohei Mitani, the *C. elegans* Gene Knockout Consortium, the *Caenorhabditis* Genetics Center provided mutant strains. We thank R Kopito and J Frydman for discussion, C Richardson, P Kurshan and members of Shen laboratory for thoughtful comments on the manuscript. We thank Caroline Yu for contribution of the screen. XW was supported by a predoctoral fellowship from the American Heart Association, Western States Affiliate (13PRE14000009).

## Additional information

### Competing interests

KS: Reviewing editor, *eLife*. The other authors declare that no competing interests exist.

### Funding

| Funder | Grant reference | Author |
| --- | --- | --- |
| National Institute of Neurological Disorders and Stroke (NINDS) | 5R01NS082208-02 | Xing Wei, Audrey S Howell, Xintong Dong, Caitlin A Taylor, Roshni C Cooper, Kang Shen |
| Howard Hughes Medical Institute (HHMI) | | Xing Wei, Audrey S Howell, Xintong Dong, Caitlin A Taylor, Roshni C Cooper, Kang Shen |
| American Heart Association (AHA) | 13PRE14000009 | Xing Wei |
| National Institutes of Health (NIH) | 1R01NS082208 | Xing Wei, Audrey S Howell, Xintong Dong, Caitlin A Taylor, Roshni C Cooper, Kang Shen |

The funders had no role in study design, data collection and interpretation, or the decision to submit the work for publication.

## Author contributions

XW, Conception and design, Acquisition of data, Analysis and interpretation of data, Drafting or revising the article; ASH, Mapped the mutants and performed the initial characterization of the mutants, Drafting or revising the article; XD, Isolated the two mutants of ire-1; CAT, Performed the initial characterization of the mutants; RCC, Wrote the Matlab codes for tracing and dividing primary dendrite; JZ, Did the non-parametric statistical analysis in R; WZ, DRS, Generated the qyls369 marker; KS, Conception and design, Drafting or revising the article

# Additional files

## Supplementary file

• Source code 1. Custom built software in Matlab and Python codes. Zip file contains: **findPrimary.m** to find the PVD primary dendrite, and it calls BDB function (Peng, H, Long, F, Liu, X, Kim, SK, and Myers, EW (2008). Straightening *Caenorhabditis elegans* images. *Bioinformatics* **24**, 234–242. Cited paper in the manuscript), which is available from http://penglab.janelia.org/proj/wormatlas/bdb_minus_demo_download.html. **SplitWorm.m** to split the primary dendrite into equal length fragments. **PrintSplits.m** to draw lines to indicate the segmentation of the primary dendrite. **ImageProcessing.py**: the python code for image processing and data analysis.

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
