## [Decision Letter]

Thank you for sending your work entitled “The Unfolded Protein Response is Required for Dendrite Morphogenesis” for consideration at *eLife*. Your article has been favorably evaluated by Randy Schekman (Senior editor) and three reviewers, one of whom is a member of our Board of Reviewing Editors.

The Reviewing editor and the other reviewers discussed their comments before we reached this decision, and the Reviewing editor has assembled the following comments to help you prepare a revised submission.

This manuscript is focused on characterizing the role that IRE1, a central component of the unfolded protein response, plays in dendrite arborization in *C. elegans*. The contribution of IRE-1 or the UPR in general to dendrite patterning has not been described, despite mounting evidence that ER homeostasis is defective in many neurodegenerative disorders. Studies in this manuscript therefore represent the first direct demonstration for a role of IRE-1 in dendrite patterning. The authors convincingly demonstrate that IRE-1 is required for dendrite patterning in PVD, and that this function likely involves protein quality control of DMA-1. While this is of general interest and the *ire-1* phenotypes are clear, there remain some issues that should be clarified.

Major concerns:

1) One weakness of the study concerns the sufficiency of *dma-1* triggered UPR in PDE morphogenesis (Figure 7). The authors show that overexpression of *dma-1* and *hsp-4* can cause abnormal shape of PDE. However, the data do not convincingly or directly show that this is due to changes in UPR or ER folding capacity. Does this ectopic expression induce changes in the XBP-1 reporter? Is the degree of ectopic branching correlating with *dma-1* protein folding or protein localization? Simply based on the data, the PDE morphological changes can also be due to signaling induced by ectopic *dma-1* expression. Since the authors find *kpc-1* and *hpo-30* are not responsible for triggering UPR in PVD, they should include ectopic expression of either gene as control for the specificity of *dma-1*.

2) DMA overexpression in IRE mutants sometimes enhances IRE phenotypes, sometimes suppresses. Can the authors provide any additional insight as to what might determine this outcome?

3) Figure 5: the induction of XBP-1::GFP based on fluorescent intensity seems subtle. The image in L4 (panel E) is hardly different from that of L3. Is this due to poor folding of GFP? The data could be strengthened by including analyses of mRNA splicing of their *xbp-1* reporter.

4) It would be appropriate and straightforward for the authors to include additional genetic and transgenic controls for the rescue experiments. Specifically, please include transgene expression alone.

5) A tighter connection to cell size seems to be necessary given that the major assertion of the paper is that the UPR is particularly required for neurons with a large dendritic tree. For example, are there genetic manipulations of cell size that are independent of UPR that might be used to suppress the sensitivity to UPR stress in these large neurons?

---

## [Author Response]

*1) One weakness of the study concerns the sufficiency of* dma-1 *triggered UPR in PDE morphogenesis (*Figure 7*). The authors show that overexpression of* dma-1 *and* hsp-4 *can cause abnormal shape of PDE. However, the data do not convincingly or directly show that this is due to changes in UPR or ER folding capacity. Does this ectopic expression induce changes in the XBP-1 reporter? Is the degree of ectopic branching correlating with* dma-1 *protein folding or protein localization? Simply based on the data, the PDE morphological changes can also be due to signaling induced by ectopic* dma-1 *expression. Since the authors find* kpc-1 *and* hpo-30 *are not responsible for triggering UPR in PVD, they should include ectopic expression of either gene as control for the specificity of* dma-1.

To answer if *dma-1* expression in PDE turns on UPR, we tested the UPR activity with the *Phsp-4::HIS-24::GFP* reporter, which serves as a readout for the transcriptional activation of the HSP-4 chaperone. We found that HIS-24::GFP level is dramatically increased in PDE neurons expressing *dma-1*, suggesting that *dma-1* expression indeed activates UPR (Figure 7—figure supplement 2). In the same experiments, we also attempted to visualize DMA-1 localization with a DMA-1::tagBFP fusion. However, unfortunately, the fluorescence of BFP protein fusion is too dim to clearly delineate the subcellular localization of DMA-1. We further tested the subcellular localization of DMA-1 and UPR activity in an attempt to address the reviewers’ next question.

As the reviewers requested, we also tested if overexpressing *hpo-30* with *hsp-4* in PDE caused ectopic branching of PVD and found no ectopic branches (Figure 7 panel F). So far, DMA-1 is the only cell surface molecule, which can induce ectopic branches.

*2) DMA overexpression in IRE mutants sometimes enhances IRE phenotypes, sometimes suppresses. Can the authors provide any additional insight as to what might determine this outcome*?

When we overexpressed *dma-1* in the *ire-1* mutants, we found that about 60% of the transgenic animals showed a “wt-like” dendrites, while about 25% of the transgenic animals showed enhanced “*dma-1-*like” dendrites. We hypothesized that the two types of phenotypes are due to variability of animals’ ability to fold overexpressed DMA-1. To directly test this hypothesis, we compared the subcellular localization of DMA-1::GFP between the “wt-like” animals and “*dma-1*-like” animals separated based on their dendrite morphology. We found that the “*dma-1*-like” animals showed 15 fold higher level of cell body DMA-1 compared with the “wt-like” animals. The correlation between intense cell body DMA-1::GFP and the enhanced dendrite phenotype is very tightly correlated. These data suggest that animals with slightly higher folding capacity can produce functional DMA-1 and rescue the dendrite morphology phenotypes. In animals with less folding capacity, this folding problem is exacerbated by overexpression of DMA-1, which leads to DMA-1 misfolding and accumulation in the ER.

*3)*
Figure 5*: the induction of XBP-1::GFP based on fluorescent intensity seems subtle. The image in L4 (panel E) is hardly different from that of L3. Is this due to poor folding of GFP? The data could be strengthened by including analyses of mRNA splicing of their* xbp-1 *reporter*.

We agree with the reviewers that the XBP-1::GFP only yielded weak fluorescence. In the revised manuscript, we verified this result by using another UPR activity reporter (*Phsp-4::HIS-24::GFP*). As an ER chaperone, HSP-4 is a transcriptional target of activated XBP-1. We reasoned that this reporter should be more sensitive due to the transcriptional amplification. Indeed, we found that the GFP signal is significant higher but also more variable between animals. Using this reporter, we repeated developmental regulation of UPR. We found that URP was activated in L3. The UPR activity was further increased in L4 animals during which the menorahs form. The UPR activity was dramatically downregulated in adult animals (Figure 5—figure supplement 1). These results are consistent with the original Figure 5.

We did consider the mRNA analysis suggested by the reviewers. This experiment will require us to isolate mRNA specifically from PVD neurons. There are only 2 PVD neurons in each worm. We would need to develop an entirely new method to collect these cells before we can attempt the experiment. Due to the time constraint of resubmission, we decided not to pursue it at this moment. However, as part of our effort to understand the transcriptome of PVD, we are in the process of adopting cell isolation method in *C. elegans*. We will definitely perform this experiment once the cell isolation system works reliably.

*4) It would be appropriate and straightforward for the authors to include additional genetic and transgenic controls for the rescue experiments. Specifically, please include transgene expression alone*.

As the reviewers requested, we now provide images for the rescuing transgenes(*PVD::ire-1* and *PVD::xbp-1 cDNA*) in the wild type background (Figure 1—figure supplement 3). We observed that neither of these transgenes altered the dendritic morphology of wt PVD. Overexpressing DMA-1 in the wt background causes overbranching of PVD dendrites and this effect was reported in [33].

*5) A tighter connection to cell size seems to be necessary given that the major assertion of the paper is that the UPR is particularly required for neurons with a large dendritic tree. For example, are there genetic manipulations of cell size that are independent of UPR that might be used to suppress the sensitivity to UPR stress in these large neurons*?

We thank the reviewers for this insightful suggestion. We tested the reviewers’ hypothesis with the *dpy-5* mutants, which have significantly reduced body size compared with wild type controls. As a result of the reduced body size, the PVD dendritic arbor is also significantly smaller in the *dpy-5* mutant compared with the wild type. We found that *dpy-5 ire-1* double mutants showed a partially suppressed the *ire-1* phenotype (Figure 7—figure supplement 1) while *dpy-5* single mutant showed normal dendrite patterns. Together, these results support the hypothesis that UPR is particularly required for neurons with large dendritic tree. By reducing the cell size, the burden of UPR is reduced accordingly.